

# Improving the cultivated rice Sakha104 (*Oryza sativa* L.) using gene pools of some relative wild species

Walid H. Elgamal[1], Mostafa M. Elshenawy[1], Samah M. Abdelkhalek[1], Dayun Tao[2], Jiawu Zhou[2], Jing Li[2] and Medhat Rehan[3]

[1] Rice Research and Training Department, Field Crops Research Institute (FCRI), Agricultural Research Center (ARC), Sakha, Egypt
[2] Yunnan Key Laboratory for Rice Genetic Improvement, Food Crops Research Institute, Yunnan Academy of Agricultural Sciences, Kunming, China
[3] Department of Plant Production, College of Agriculture and Food, Qassim University, Buraydah, Saudi Arabia

Corresponding author
Medhat Rehan, m.rehan@qu.edu.sa

## ABSTRACT

**Background:** Rice (*Oryza sativa* L.) is considered a staple food for one-half of the world's population. The yield of rice must increase to keep up with the world's population growth. Blast disease (caused by *Magnaporthe oryzae*) is biotic stress that threatens rice production and can result in yield losses up to 70%.

**Methods:** The present research attempted to widen the genetic base of Egyptian cultivated rice Sakha 104 (*Oryza sativa*), using gene pools from certain relative wild ancestors, in order to cope with blast infection and grain yield. Crossing Sakha 104 × *O. glaberrima* and Sakha 104 × *O. glumaepatula* resulted in selecting 20 genotypes. The produced genotypes and the Egyptian cultivar Sakha 104 were assessed for days to heading (HD), plant height (PH), number of tillers/plant (NTP), panicle weight (PW), 1,000-grain weight (TGW), grain yield/plant (GYP), spikelet fertility (SF), blast reaction (BR), hulling percentage (HP), milling percentage (MP), head rice (HR), and amylose content (AC).

**Results:** Line AS-AF L3 had the earliest heading date, whereas AS-AF L6 revealed the lowest and the best values in plant height. In addition, line AS-AM L9 generated the most tillers/plant and the heaviest panicle weight. For TGW, AS-AM L3 showed the uppermost value, while AS-AM L4 recorded the highest percentage in spikelet fertility and high productivity of grain yield/plant. Furthermore, all assessed genotypes presented a unity (the value of 1) across the two seasons of evaluation in blast reaction. Grain quality criteria such as hulling, milling percentages and head rice assigned to AS-AF L10 and AS-AM L3, whereas AS-AF L2 possessed the lowest values in amylose content. Moreover, genetic variance (GV), phenotypic variance (PV), genotypic and phenotypic coefficient variations (GCV and PCV) were estimated for all traits with higher PV and PCV than GV and GCV, respectively. Heritability in broad sense (h2b%) disclosed high heritability values for heading date (0.85), plant height (0.925), grain yield/plant (0.95), 1,000 grains weight (0.92), blast reaction (0.935), head rice (0.97) and amylose content (0.90), reflecting strong genetic control of these traits. Eventually, broadening the genetic background of Sakha 104 cultivar against blast infection will minimize its impact and enhance the food security in Egypt.

# INTRODUCTION

More than 3.5 billion people rely on rice for more than 20% of their daily calories' intake. Global rice demand is predicted to expand from 763 million tons in 2020 to 852 million tons in 2035, with an overall increase of 18% in the following two decades (*Brar & Khush, 2018*). Global rice production has more than doubled during the last 40 years. This was mostly accomplished using the concepts of classical Mendelian genetics and traditional plant breeding (*Brar & Singh, 2011*). Rice productivity is continually threatened by several biotic, abiotic stresses and climatic changes. To overcome these constraints, there is an urgent need to broaden the gene pool of rice cultivars; one of the options is to exploit wild species of *Oryza* which are reservoirs of useful genes or quantitative trait loci (QTLs) for rice development (*Surapaneni et al., 2017*). Wild rice species are major sources of economically relevant traits, such as tolerance to acid soils and drought, yield potential, cytoplasmic male-sterility, heat tolerance, and elongation ability (*Henry, 2022*). For instance, *Oryza rufipogon* owns acid soil tolerance trait (*Mandal & Gupta, 1997*), *O. granulate* and *O. meyeriana* have low light intensity tolerance (*Vaughan, Morishima & Kadowaki, 2003*), whereas *O. nivara* possesses high micronutrient content (*Swamy et al., 2012*; *Gaikwad et al., 2021*). These features may be exploited through breeding programs by developing large-scale of genetic resources (*Zhang et al., 2022a*).

Every year, the fungal pathogen *Magnaporthe grisea* (*Pyricularia oryzae*) causes the rice blast (biotic stress) and destroys a huge area from cultivated rice that is enough to feed more than 60 million people (*Khush & Jena, 2009*). The fungus attacks rice plants (aboveground tissues) at any development stage and produces lesions on leaves (leaf blast), panicle and panicle neck nodes (panicle blast and neck rot), leaf collars (collar blast), in addition to culms and culm nodes. These symptoms vary in shape and color based on the plant age, climatic circumstances and variety resistance (*Asibi, Chai & Coulter, 2019*). The breakdown of blast resistance in rice varieties led to instability in rice yield over the world. In addition, developing durable blast resistance in rice cultivars is highly desirable to cope with the huge demand on rice production. QTL genes for blast resistance are necessary to be identified and transferred to new durably rice varieties from the wild rice ancestors. Therefore, blast-resistance (*R*) genes with broad-spectrum and high breeding value consider the most attractive, economical and effective approach to produce resistant cultivars and control rice blast (*Khush & Jena, 2009*). More than 100 qualitative resistance (*R*) genes and 350 quantitative resistance loci (QRLs) were identified and conferred resistance to blast (*Yin et al., 2021*; *Feng et al., 2022*; *Tian et al., 2022*).

*Oryza glumaepatula* is the only native wild rice from the Americas that has a diploid genome ($2n = 24$). Thus, these species constitute a source for genetic variability to enhance *Oryza sativa* cultivars (*Pegoraro, da Rosa Farias & de Oliveira, 2018*). The transference of alleles associated to target characters from *O. glumaepatula* into *O. sativa* could be achieved *via* introgression or using genetic engineering. *O. glumaepatula* occurs in central

and South America and Caribe, growing in flooded areas, swamps, rivers, and humid areas with clay soils that present invasive or colonizing behavior. Recently, *O. glumaepatula* had its genome sequenced, enabling comparative studies among different *Oryza* species (*Ejiri, Sawazaki & Shiono, 2020*). The expansion and contraction of gene families, diversity, variation in the number of noncoding genes and divergence in the sequences among the species are associated with the adaptation in each species with different environmental conditions (*Sánchez & Espinoza, 2005*).

*Oryza glaberrima* is an African cultivated rice variety that was domesticated from its wild ancestor by farmers in the Niger River's inland delta. Several investigations revealed that it has extremely restricted genetic diversity relative to its wild ancestor (*Ndjiondjop et al., 2018*). *O. glaberrima* has significant value not only in Africa but also globally. It makes a great contribution to regional and global food security as a source of genes that possess resistance/tolerance to various biotic and abiotic stressors. It also has distinctive starch-related characteristics that give it exceptional cooking and eating features (*Kang et al., 2008*). Additionally, advances in DNA sequencing have offered significant genetic resources for African rice, particularly whole genome sequences. Genomic tools are enabling greater understanding of the useful functional diversity found in this species (*Wambugu, Ndjiondjop & Henry, 2021*). These developments have the ability to resolve some of the unfavorable characteristics observed in this species, which has led to its continued displacement by Asian rice. The development of a new generation of rice varieties for African farmers would necessitate the use of modern molecular breeding technologies that will enable efficient application of the riches and resilience inherent in African rice to enhance rice programs (*Sarla & Swamy, 2005*).

Sakha 104 has been regarded as one of the most frequent and widely spread cultivars (20%) in Egypt over the past two decades. It was released in 1999 as a typical japonica plant type generated from the local cross GZ 4096-8-l/GZ 4100-9-1, with excellent yield potential and moderate salt tolerance (*Rice Research and Training Center, 2022*). Furthermore, it had resistance to a variety of diseases (including blast and brown spots) and insects (such as stem borers). Additionally, it possesses short grains with high milling percentage (73%) coupled with good cooking and eating properties (*Elmoghazy & Elshenawy, 2019*). During the 2006 growing season, Sakha 104's blast resistance had deteriorated (*Sehly et al., 2008*) and represented less area. Due to the importance of *Oryza* wild relative's species, this investigation aimed at: (i) Improving *Oryza sativa* cultivated rice (Sakha 104) using useful wild genotypes and gene pool, (ii) broaden the genetic base of Egyptian cultivated rice, (iii) generating new rice genotypes with blast resistance to secure rice yielding, and (iv) assessing the related-yield traits in new generated genotypes under blast infection.

## MATERIALS AND METHODS

### Experiential design and plant materials

The present study was performed at the Experimental Farm of rice research department, in Sakha, Kafr El-Sheikh, Egypt (31.0894°N, 30.9444°E). A total of 21 genotypes involving 20 promising lines derived from two separate crosses, the 1st cross

(Sakha 104 × *O. glaberrima*), while the 2nd cross (Sakha 104 × *O. glumaepatula*), as well as the Egyptian background of Sakha 104 cultivar (*Oryza sativa* L.) were generated and assessed. Two accessions of *O. glaberrima* (IRGC101901) and *O. glumaepatula* (Acc.100184) were crossed with the Chinese variety Dianjingyou1 (DJY1, bred by Yunnan Academy of Agricultural sciences (YAAS)), to develop introgression lines (ILs) by back crossing. Based on phenotypic evaluation, two selected introgression lines (ILs) were used as donor to pass the wild relatives genes pool to the Egyptian cultivar (Sakha 104) by crossing to produce F1 generation. The selection based on phenotyping started from F2 (from 1,800 single plants came from the 1st cross and 2,000 single plants obtained from the 2nd cross). A 33 and 37 F3 families (resulted from 33 and 37 F2 single plants) represented the F3 generation from the two crosses, respectively. In F4 generation, 28 and 29 single plants from best families were selected and reduced to 21 and 25 plants in F5, respectively. The selection continued and reached 15 and 18 lines in F6, then, the best stable lines (8 and 12 lines, respectively) were chosen for F7 and F8 for the final evaluation.

The crossing and early generations were implemented at Yunnan province, China, whereas the late generations and final evaluation were achieved in Egypt as presented in Fig. 1. The promising lines with the cultivar Sakha 104 were evaluated under normal field and blast nursery conditions in independent trials using randomized complete block design (RCBD) in three replicates for two summer growing seasons (2021 and 2022). The first experiment conducted routinely, and all recommended applications were followed in accordance with RRTC instructions during the crop growth period to achieve an optimal crop stand. Thirty-day old seedlings were transported for transplanting and each genotype planted in three rows per replication. The row was five meters long with 20 × 20 cm spacing between plants. The data collected and analyzed based on the Standard Evaluation System of the *International Rice Research Institute (IRRI) (2013)* under both normal and blast circumstances.

### Evaluation of the studied genotypes against blast disease

In the second field experiment with open field stress conditions (high plant density, high fertilization and late season sowing), 21 rice genotypes (Sakha 104 as a susceptible checks and 20 derived promising lines), were assessed for seedling reaction under blast nursery (natural conditions) at Sakha station, Kafr El-Sheikh governorate with three replications for each. The seedbed was prepared by adding manure fertilizer during land preparation (20 $m^3$/ha). Each entry was planted in three rows by direct seeding (each row was 50 cm long and 15 cm apart). The 15th of June was the sowing date for the 2021 and 2022 crop seasons. Natural infection was developed, and plants were scored for leaf infection at 40–50 days after sowing. The SES of 0–9 scale designed by *International Rice Research Institute (IRRI) (2013)* was used as follows: leaf blast score, 1–2 = resistant (R), 3 = moderately resistant (MR), 4–6 = susceptible (S), 7–9 = highly susceptible (HS).

### Data collection

Data were recorded on a random 10 plants from each center row per replicate. The investigated parameters were, days to heading (HD, day), plant height (PH, cm), number

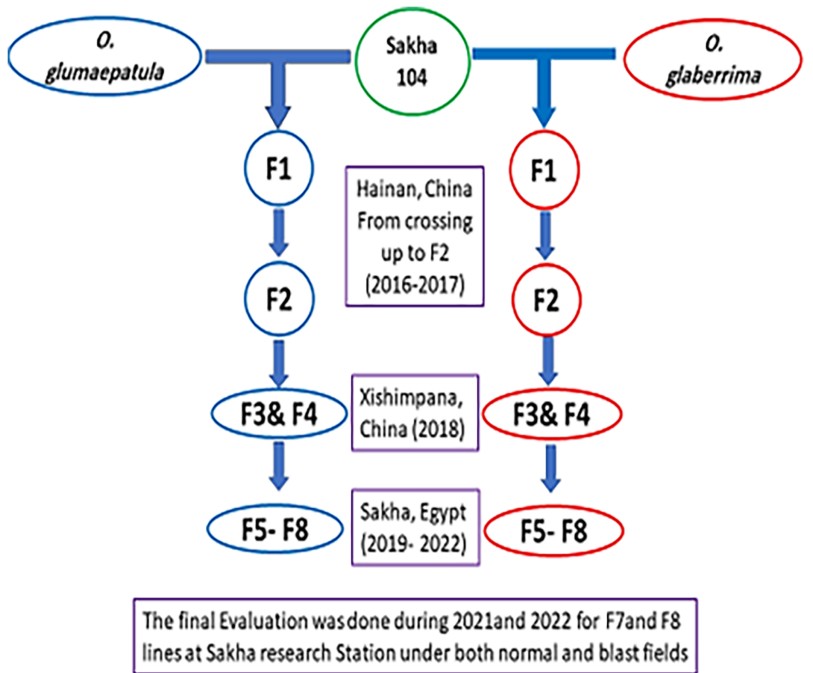

Figure 1 The development of new derived lines from Sakha 104 and its genetic background.

of tillers/plant (NTP), panicle weight (PW, g), 1,000 grains weight (TGW, g), grain yield/plant (GYP, g), spikelet fertility (SF, %) and blast reaction (BR, %). The grain quality features included hulling percentage (HP, %), milling percentage (MP, %), head rice (HR, %), and amylose content (AC, %).

## Data analysis

The analysis of variance for each experiment (normal and blast conditions) was done according to *Steel & Torrie (1962)*. The analysis software was Genstat for Windows (nineteenth edition; VSN International Ltd., Hemel Hempstead, UK). The genetic parameters were computed according to formula suggested by *Burton (1952)* and *Hanson, Robinson & Comstock (1956)*.

## RESULTS

### Analysis of variance

The analysis of variance is very important for the breeders as well as sources of variance. The mean square estimates highly significant values between genotypes for all assessed traits that confirmed by growing seasons 2021 and 2022 as presented in Tables 1 and 2. These significant differences in heading date (HD), plant height (PH), number of tillers/plant (NTP), blast reaction (BR), panicle weight (PW), grain yield/plant (GYP), 1,000 grains weight (TGW), spikelet fertility (SF), hulling percentage (HP), milling percentage (MP), head rice (HR), and amylose content (AC), reflect real differences among the studied genotypes produced by inter specific crossing and produced a broaden genetic base.

**Table 1 Mean squares and analysis of variance for some agronomic, yield and some yield components traits as well as blast reaction over two growing seasons.**

| SOV | df | Year | HD (day) | PH (cm) | NTP | BR | PW (g) | GYP (g) |
|---|---|---|---|---|---|---|---|---|
| Genotypes | 20 | 2021 | 25.787** | 296.26** | 55.020** | 1.285** | 0.932** | 271.10** |
| | | 2022 | 29.130** | 268.06** | 52.021** | 1.587** | 0.738** | 276.92** |
| Replicates | 2 | 2021 | 3.444 | 4.111 | 1.710 | 0.048 | 0.159 | 8.579 |
| | | 2022 | 1.016 | 1.444 | 4.778 | 0.016 | 0.048 | 9.761 |
| Error | 40 | 2021 | 1.861 | 9.628 | 10.800 | 0.047 | 0.082 | 4.192 |
| | | 2022 | 0.999 | 5.956 | 5.911 | 0.016 | 0.069 | 4.605 |

Notes:
**Significant differences at 0.01 level of probability.
SOV, source of variance; df, degree of freedom; HD, heading date; PH, plant height; NTP, number of tillers/plant; PW, panicle weight; GYP, grain yield/plant.

**Table 2 Mean squares and analysis of variance model for 1,000 grains weight, fertility percentage and some studied grain quality traits over two growing seasons.**

| SOV | df | Year | 1,000-GW (g) | SF (%) | HP (%) | MP (%) | HR (%) | AC (%) |
|---|---|---|---|---|---|---|---|---|
| Genotypes | 20 | 2021 | 20.171** | 51.860** | 7.9289** | 9.366** | 179.06** | 20.756** |
| | | 2022 | 20.920** | 51.421** | 7.6717** | 8.877** | 168.62** | 14.501** |
| Replicates | 2 | 2021 | 1.649 | 1.437 | 0.86375 | 0.659 | 1.070 | 0.872 |
| | | 2022 | 1.062 | 3.044 | 1.10392 | 0.122 | 0.436 | 4.911 |
| Error | 40 | 2021 | 0.817 | 3.037 | 0.40738 | 0.912 | 1.041 | 0.333 |
| | | 2022 | 0.339 | 1.756 | 0.78546 | 0.210 | 2.029 | 0.826 |

Notes:
**Significant differences at 0.01 level of probability.
1,000-GW, 1,000-grains weight; SOV, source of variance; df, degree of freedom; SF, Spikelet fertility; HP, hulling percentage; MP, milling percentage; HR, head rice; AC, amylose content.

## Mean performances

The evaluated 21 rice genotypes (20 promising lines generated from crosses among Sakha 104 × *Oryza glaberrima* and Sakha 104 × *O. glumaepatula* in addition to the Egyptian back ground cultivar (Sakha 104, Oryza *sativa* L)) were evaluated for some morphological, physiological and yield-related traits. The mean values presented in Tables 3, 4 and 5, displayed that no line was superior in all studied traits but there was exchange in superiority observed among the evaluated traits. For heading date and plant height (Table 3), the lowest values are preferred supporting the early maturing and semi dwarfism for lodging resistance. Lines AS-AF L3, AS-AF L6, AS-AM L1 and AS-AM L3 showed the earliest heading performance reached 11.55%, 9.97%, 9.8% and 9.97% when compared with the variety Check (sakha 104), respectively, between all promising lines for the two growing seasons. AS-AF L6, AS-AF L3, AS-AM L8 and AS-AM L1 exhibited the lowest and the best values that recorded 23.3%, 18%, 19% and 14.7% decrease in plant height, respectively. Furthermore, lines AS-AM L9, AS-AM L6 and AS-AF L2 produced the maximum number of tillers/plant with average 31.84, 31.34 and 30.5, respectively, whereas the best panicle weight was found in lines AS-AM L9, AS-AML7 and AS-AML4 with an enhancement recorded 39%, 32.8% and 32%, in the same order, than sakha 104.

**Table 3 Heading date, plant height, number of tillers/plant and panicle weight for the assessed rice genotypes during 2021 and 2022 growing seasons.**

| Genotypes | HD (days) | | PH (cm) | | NTP | | PW (g) | |
|---|---|---|---|---|---|---|---|---|
| Season | 2021 | 2022 | 2021 | 2022 | 2021 | 2022 | 2021 | 2022 |
| AS-AF L1 | 97.33 | 96.00 | 113.33 | 112.67 | 26.00 | 25.33 | 4.66 | 4.17 |
| AS-AF L2 | 101.33 | 100.33 | 101.33 | 102.00 | 30.67 | 30.33 | 4.36 | 4.20 |
| AS-AF L3 | 94.33 | 92.00 | 95.33 | 98.00 | 21.33 | 20.67 | 3.80 | 3.47 |
| AS-AF L4 | 99.00 | 99.33 | 121.00 | 118.17 | 22.67 | 22.33 | 3.81 | 3.39 |
| AS-AF L5 | 99.67 | 99.67 | 120.33 | 119.67 | 26.67 | 26.67 | 4.12 | 4.17 |
| AS-AF L6 | 95.33 | 94.33 | 91.33 | 89.67 | 22.33 | 22.67 | 4.70 | 4.50 |
| AS-AF L7 | 100.00 | 100.33 | 108.00 | 107.00 | 25.67 | 26.67 | 4.95 | 5.12 |
| AS-AF L8 | 102.00 | 100.00 | 110.67 | 109.17 | 28.00 | 29.67 | 5.00 | 5.07 |
| AS-AM L1 | 95.33 | 94.67 | 100.67 | 100.67 | 22.00 | 21.33 | 4.55 | 4.68 |
| AS-AM L2 | 99.67 | 99.67 | 106.67 | 104.67 | 17.33 | 17.67 | 4.30 | 4.19 |
| AS-AM L3 | 95.67 | 94.00 | 104.00 | 104.33 | 25.00 | 24.67 | 4.37 | 4.40 |
| AS-AM L4 | 98.33 | 97.67 | 124.00 | 120.67 | 24.67 | 24.33 | 5.28 | 5.03 |
| AS-AM L5 | 97.67 | 96.00 | 113.67 | 113.33 | 18.67 | 19.00 | 5.23 | 4.92 |
| AS-AM L6 | 97.67 | 98.00 | 123.00 | 122.67 | 31.67 | 31.00 | 4.68 | 4.46 |
| AS-AM L7 | 102.33 | 99.67 | 106.33 | 106.33 | 26.00 | 25.33 | 5.29 | 5.08 |
| AS-AM L8 | 97.33 | 96.67 | 96.00 | 95.00 | 22.33 | 22.67 | 4.37 | 4.43 |
| AS-AM L9 | 103.67 | 102.00 | 124.67 | 123.00 | 32.00 | 31.67 | 5.37 | 5.49 |
| AS-AF L10 | 102.67 | 101.67 | 103.33 | 103.00 | 28.33 | 27.33 | 4.99 | 5.14 |
| AS-AM L11 | 98.33 | 98.00 | 105.33 | 103.67 | 28.33 | 28.33 | 5.15 | 4.82 |
| AS-AM L12 | 102.00 | 100.67 | 115.33 | 114.67 | 24.33 | 24.00 | 4.51 | 4.87 |
| Sakha104 | 105.67 | 105.00 | 118.00 | 118.00 | 17.00 | 17.67 | 3.89 | 3.92 |
| LSD 5% | 2.328 | 2.343 | 2.631 | 2.618 | 2.419 | 2.412 | 1.972 | 1.949 |

**Note:**
AS-AFL (1–8), Asian-African lines (1–8); AS-AML (1–12), Asian-American lines (1–12); HD, heading date; PH, plant height; NTP, number of tillers/plant; PW, panicle weight; LSD, least significant difference.

**Table 4 Thousand grains weight, spikelet fertility, grain yield/plant and blast reaction for studied rice genotypes during 2021 and 2022 growing seasons.**

| Genotypes | 1,000-GW (g) | | SF (%) | | GYP (g) | | BR | |
|---|---|---|---|---|---|---|---|---|
| Season | 2021 | 2022 | 2021 | 2022 | 2021 | 2022 | 2021 | 2022 |
| AS-AF L1 | 30.30 | 30.45 | 91.07 | 90.41 | 52.42 | 51.60 | 1.00 | 1.00 |
| AS-AF L2 | 29.70 | 29.77 | 82.41 | 81.77 | 61.82 | 57.47 | 1.00 | 1.00 |
| AS-AF L3 | 29.43 | 29.43 | 93.70 | 94.34 | 53.39 | 52.92 | 1.00 | 1.00 |
| AS-AF L4 | 24.23 | 24.30 | 84.99 | 85.38 | 57.07 | 57.30 | 1.00 | 1.00 |
| AS-AF L5 | 21.85 | 21.75 | 88.74 | 88.78 | 64.45 | 62.63 | 1.00 | 1.00 |
| AS-AF L6 | 24.42 | 24.42 | 92.65 | 93.53 | 49.12 | 47.50 | 1.00 | 1.00 |
| AS-AF L7 | 25.40 | 25.47 | 94.52 | 94.53 | 63.45 | 64.25 | 1.00 | 1.00 |
| AS-AF L8 | 28.07 | 28.20 | 92.04 | 91.44 | 63.81 | 60.17 | 1.00 | 1.00 |
| AS-AM L1 | 23.77 | 24.23 | 89.47 | 88.86 | 53.10 | 52.73 | 1.00 | 1.00 |

| Genotypes | 1,000-GW (g) | | SF (%) | | GYP (g) | | BR | |
|---|---|---|---|---|---|---|---|---|
| Season | 2021 | 2022 | 2021 | 2022 | 2021 | 2022 | 2021 | 2022 |
| AS-AM L2 | 23.89 | 23.89 | 86.29 | 86.25 | 56.81 | 56.00 | 1.00 | 1.00 |
| AS-AM L3 | 30.80 | 30.93 | 94.17 | 93.81 | 55.59 | 54.04 | 1.00 | 1.00 |
| AS-AM L4 | 28.66 | 29.09 | 94.84 | 94.77 | 68.73 | 67.77 | 1.00 | 1.00 |
| AS-AM L5 | 25.13 | 25.33 | 94.48 | 93.25 | 52.83 | 51.61 | 1.00 | 1.00 |
| AS-AM L6 | 26.75 | 25.75 | 92.69 | 93.51 | 51.66 | 59.07 | 1.00 | 1.00 |
| AS-AM L7 | 30.53 | 30.80 | 92.93 | 92.77 | 60.30 | 59.31 | 1.00 | 1.00 |
| AS-AM L8 | 29.10 | 29.40 | 92.99 | 93.21 | 62.59 | 61.59 | 1.00 | 1.00 |
| AS-AM L9 | 26.89 | 26.56 | 89.69 | 90.09 | 60.77 | 64.60 | 1.00 | 1.00 |
| AS-AF L10 | 25.90 | 26.17 | 85.48 | 85.30 | 64.47 | 68.04 | 1.00 | 1.00 |
| AS-AM L11 | 27.55 | 27.55 | 93.11 | 92.21 | 61.08 | 60.41 | 1.00 | 1.00 |
| AS-AM L12 | 24.83 | 25.14 | 93.96 | 93.62 | 64.06 | 68.26 | 1.00 | 1.00 |
| Sakha 104 | 28.37 | 28.53 | 95.15 | 95.83 | 45.96 | 45.06 | 4.00 | 4.00 |
| LSD 5% | 2.668 | 2.660 | 2.412 | 2.410 | 2.619 | 2.622 | 2.004 | 2.026 |

**Note:**
AS-AFL (1–8), Asian-African lines (1–8); AS-AML (1–12), Asian-American lines (1–12). 1,000-GW, 1,000-grains weight; SF, Spikelet fertility; GYP, grain yield/plant; BR, blast reaction; LSD, least significant difference.

**Table 5 Hulling and milling percentages, head rice and amylose content for rice genotypes under study during two growing seasons (2021 and 2022).**

| Genotypes | HP (%) | | MP (%) | | HR (%) | | AC (%) | |
|---|---|---|---|---|---|---|---|---|
| Season | 2021 | 2022 | 2021 | 2022 | 2021 | 2022 | 2021 | 2022 |
| AS-AF L1 | 80.57 | 79.86 | 71.78 | 70.98 | 50.43 | 51.14 | 29.00 | 28.53 |
| AS-AF L2 | 76.22 | 76.07 | 67.20 | 67.53 | 56.30 | 56.60 | 19.24 | 18.83 |
| AS-AF L3 | 80.67 | 79.78 | 72.04 | 71.47 | 60.45 | 61.15 | 22.69 | 22.80 |
| AS-AF L4 | 80.91 | 80.38 | 69.67 | 69.77 | 42.00 | 42.67 | 24.31 | 24.21 |
| AS-AF L5 | 82.74 | 82.80 | 70.55 | 70.52 | 40.42 | 41.14 | 25.59 | 25.40 |
| AS-AF L6 | 82.74 | 82.21 | 73.56 | 73.60 | 55.67 | 55.89 | 20.40 | 20.93 |
| AS-AF L7 | 79.90 | 79.48 | 73.56 | 73.27 | 50.71 | 51.07 | 25.99 | 25.62 |
| AS-AF L8 | 79.95 | 79.78 | 73.35 | 73.45 | 50.52 | 50.77 | 21.60 | 21.07 |
| AS-AM L1 | 80.48 | 79.83 | 70.66 | 70.63 | 41.34 | 41.45 | 27.33 | 27.11 |
| AS-AM L2 | 80.52 | 80.69 | 73.41 | 73.70 | 49.75 | 50.58 | 23.73 | 23.82 |
| AS-AM L3 | 82.93 | 82.98 | 74.01 | 73.96 | 70.00 | 69.67 | 22.29 | 21.86 |
| AS-AM L4 | 83.00 | 82.44 | 70.00 | 70.00 | 60.00 | 59.29 | 22.18 | 21.99 |
| AS-AM L5 | 80.60 | 80.53 | 70.67 | 70.56 | 52.37 | 53.46 | 22.11 | 21.74 |
| AS-AM L6 | 80.00 | 79.67 | 70.18 | 70.50 | 51.31 | 52.10 | 21.31 | 20.87 |
| AS-AML7 | 82.55 | 81.88 | 72.84 | 72.61 | 62.21 | 62.85 | 21.61 | 21.74 |
| AS-AM L8 | 80.00 | 80.02 | 73.28 | 73.20 | 63.47 | 63.38 | 23.38 | 23.59 |
| AS-AM L9 | 79.85 | 79.24 | 70.41 | 70.50 | 50.50 | 50.50 | 24.35 | 24.23 |
| AS-AF L10 | 83.46 | 82.82 | 72.94 | 72.90 | 64.11 | 64.22 | 23.14 | 23.02 |
| AS-AML11 | 80.29 | 81.10 | 70.28 | 70.65 | 55.85 | 55.32 | 25.83 | 25.42 |

| Table 5 (continued) | | | | | | | | |
|---|---|---|---|---|---|---|---|---|
| Genotypes | HP (%) | | MP (%) | | HR (%) | | AC (%) | |
| Season | 2021 | 2022 | 2021 | 2022 | 2021 | 2022 | 2021 | 2022 |
| AS-AM L12 | 80.52 | 80.67 | 73.23 | 73.58 | 56.44 | 57.15 | 23.43 | 23.15 |
| Sakha104 | 81.75 | 81.33 | 71.94 | 70.93 | 61.6 | 59.5 | 17.83 | 18.07 |
| LSD 5% | 2.195 | 2.192 | 2.214 | 2.208 | 2.566 | 2.558 | 2.304 | 2.263 |

**Note:**
AS–AFL (1–8), Asian-African lines (1–8); AS-AML (1–12), Asian-American lines (1–12); HP, hulling percentage; MP, milling percentage; HR, head rice; AC, amylose content; LSD, least significant difference.

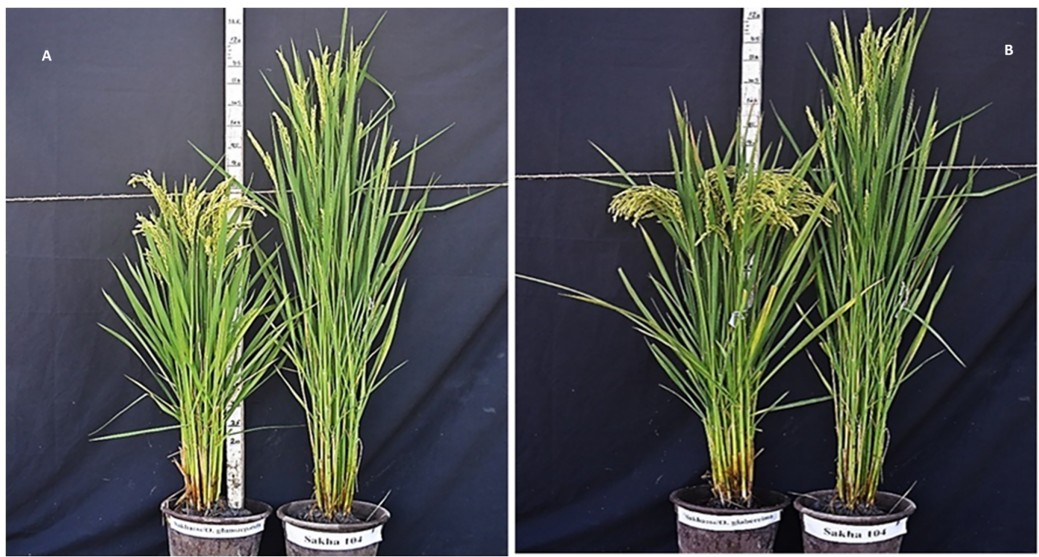

**Figure 2 The morphological differences between the new derived lines.** From (A) *O. glumaepatula* (AS-AM L8) and (B) *O. glaberrima* (AS-AF L3) with the genetic background (Sakha 104).

Likewise, Fig. 2 represents the morphological differences among the new generated lines from *O. glumaepatula* × Sakha 104 and *O. glaberrima* × Sakha 104. Comparing the days to heading and plant hight among Sakha 104 and the obtained maximum line, minimum line and the lines average, reflects decrease in these traits over the 2 years (Figs. 3A and 3B). On the other hand, number of tillers/plant and grain yield/plant displayed opposite trend with an increment in these characteristics in the maximum line and lines average compared to Sakha 104 (Figs. 3C and 3D).

The resulted 20 promising lines and their background parent Sakha 104 evaluated for 1,000-grains weight, spikelet fertility, grain yield/plant and blast reaction. The uppermost values were given by lines AS-AM L3, AS-AM L7 and AS-AF L1 with an increase counted 8.5%, 7.8% and 6.8% for TGW, respectively, while Sakha 104, AS-AM L4 and AS-AF L7 recorded the best percentage in spikelet fertility (Table 4). Grain yield/plant increased significantly in the promising lines and the best values were observed in lines AS-AM L4, AS-AF L10, AS-AM L12 and AS-AF L7 with percentages attained 50, 45.6, 45.4 and 40.3,

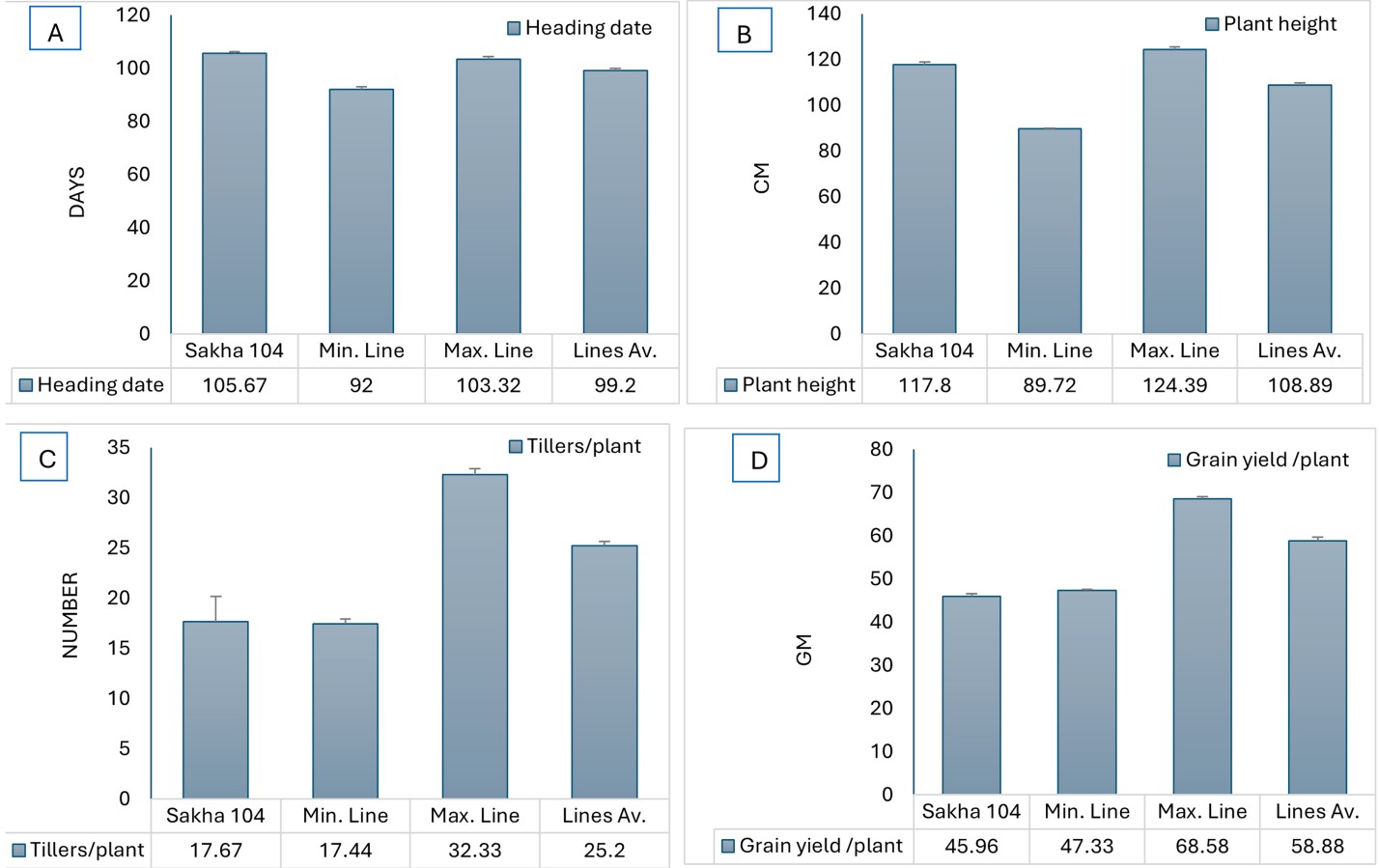

**Figure 3  Comparison of phenotype between Sakha 104 and new derived lines, from left to right.** (A) Heading date, (B) plant height, (C) tillers/plant, and (D) grain yield/plant. Sakha 104 is the Egyptian background cultivar, Min. line is the minimum line value, Max. line is the maximum line value, and the lines Av. is the average values between all lines over 2 years.

respectively, whereas the value of leaf blast reaction for all evaluated lines showed a unity across the two seasons of estimation.

When evaluating the hulling and milling percentages, head rice and amylose content, AS-AF L10, AS-AM L3, AS-AF L5, AS-AML4 and AS-AF L6 genotypes had the topmost percentages in hulling, whereas genotypes AS-AM L3, AS-AF L6, AS-AF L7, AS-AF L8, AS-AM L2, AS-AM L8 and AS-AM L12 exhibited the maximum milling percentage (Table 5). For head rice, AS-AM L3, AS-AF L10 and AS-AM L8 possessed the highest values in the characterized genotypes, while the least values among the derived lines in amylose content were detected in AS-AF L2 and AS-AF L6 genotypes since low amylose content is a desirable trait.

The constructed genotypes were examined for blast reaction and grain quality, subsequently, the maximum and minmum genotypes in addition to the genotypes average were compared to the original cultivar (Sakha 104) and presented in Fig. 4. The produced genotypes showed the lowest values in blast reaction compared to Sakha 104, reflecting high resistance to leave blast infection which is a desirable character (Fig. 4B). Otherwise,

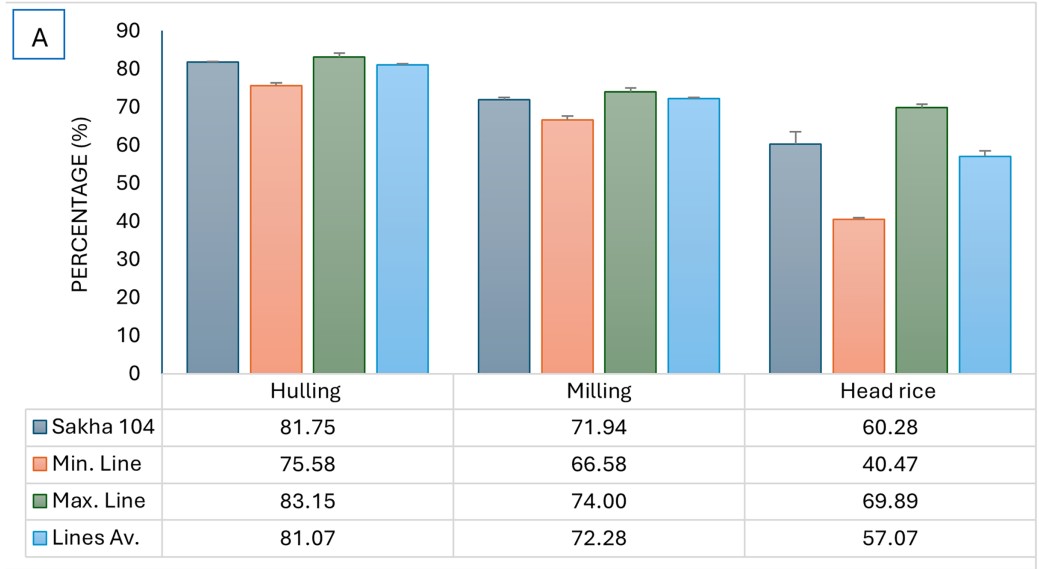

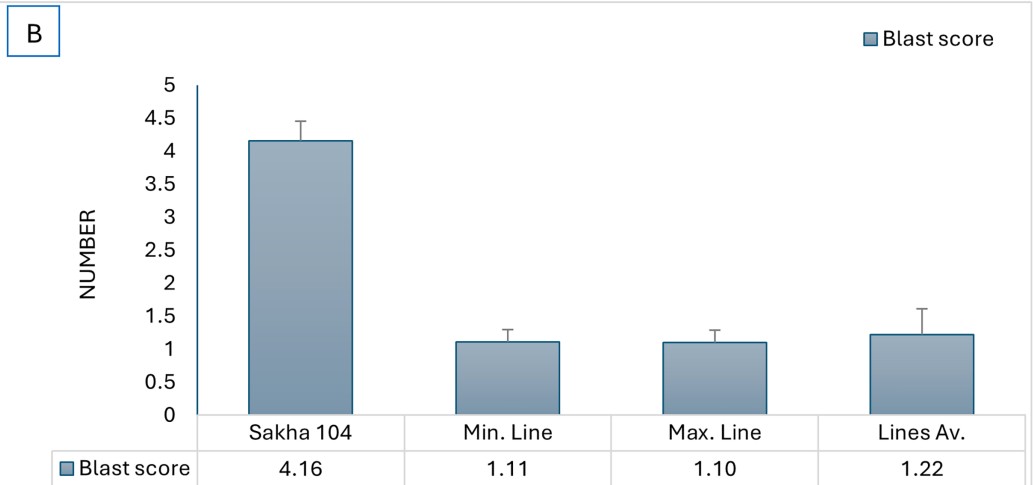

**Figure 4 Comparison of phenotype between Sakha 104 and the new derived lines, from left to right.** (A) Hulling grains percentage, milling grains percentage, and head rice, (B) blast reaction score. Sakha 104 is the Egyptian background cultivar, Min. line is the minimum line value, Max. line is the maximum line value, and the lines Av. is the average value among all evaluated lines over 2 years.

the average grain quality characterstics expressed the same as in the parent (Sakha 104) (Fig. 4A).

Analysis of variance for all estimated triats in the derived genotypes implied significant impacts of genetic variance (GV), phenotypic variance (PV), genotypic and phenotypic coefficient variations (GCV and PCV) (Table 6). In all studied traits, the PV had higher values than GV indicating significant impact of environment on theses traits. The uppermost genotypic and phenotypic coefficient variations (GCV and PCV) were detected in blast reaction trait (68.3 and 70.6), followed by NTP (15.44% and 19.22%) and GYP (15.5% and 15.86%). In addition, PCV explored higher values than GCV suggesting more

**Table 6 Variance components, estimates of phenotypic coefficient of variation (PCV) and genotypic coefficient of variation (GCV), heritability in broad sense (*h2b*%) and genetic advance (GA) for the estimated traits during 2021and 2022 growing seasons.**

| Traits | Seasons | Variance | | Coefficient of variance | | h2b | GA | GA% |
|---|---|---|---|---|---|---|---|---|
| | | GV | PV | GCV | PCV | | | |
| HD | 2021 | 7.98 | 9.84 | 2.84 | 3.16 | 0.81 | 16.43 | 16.53 |
| | 2022 | 9.38 | 10.38 | 3.11 | 3.27 | 0.90 | 19.32 | 19.64 |
| PH | 2021 | 95.55 | 105.17 | 8.92 | 9.35 | 0.91 | 196.82 | 179.53 |
| | 2022 | 87.37 | 93.32 | 8.59 | 8.87 | 0.94 | 179.98 | 165.31 |
| NTP | 2021 | 14.74 | 25.54 | 15.24 | 20.05 | 0.58 | 30.36 | 120.49 |
| | 2022 | 15.37 | 21.28 | 15.63 | 18.39 | 0.72 | 31.66 | 126.23 |
| Blast reaction | 2021 | 0.41 | 0.46 | 64.24 | 67.80 | 0.90 | 0.850 | 85.01 |
| | 2022 | 0.52 | 0.54 | 72.37 | 73.46 | 0.97 | 1.079 | 107.90 |
| PW | 2021 | 0.283 | 0.365 | 11.472 | 13.026 | 0.776 | 0.584 | 12.58 |
| | 2022 | 0.223 | 0.293 | 10.371 | 11.890 | 0.761 | 0.459 | 10.08 |
| GYP | 2021 | 88.969 | 93.161 | 15.362 | 15.720 | 0.955 | 183.28 | 298.50 |
| | 2022 | 90.772 | 95.377 | 15.616 | 16.007 | 0.952 | 186.99 | 306.49 |
| 1,000-GW | 2021 | 6.45 | 7.27 | 9.46 | 10.04 | 0.89 | 13.29 | 49.48 |
| | 2022 | 6.86 | 7.20 | 9.73 | 9.96 | 0.95 | 14.13 | 52.47 |
| Fertility % | 2021 | 16.27 | 19.31 | 4.40 | 4.80 | 0.84 | 33.53 | 36.60 |
| | 2022 | 16.56 | 18.31 | 4.45 | 4.68 | 0.90 | 34.10 | 37.32 |
| Hulling % | 2021 | 2.51 | 2.91 | 1.96 | 2.11 | 0.86 | 5.165 | 6.381 |
| | 2022 | 2.30 | 3.08 | 1.88 | 2.18 | 0.75 | 4.729 | 5.863 |
| Milling % | 2021 | 2.82 | 3.73 | 2.34 | 2.69 | 0.76 | 5.805 | 8.098 |
| | 2022 | 2.89 | 3.10 | 2.37 | 2.46 | 0.93 | 5.952 | 8.309 |
| Head rice % | 2021 | 59.34 | 60.38 | 14.12 | 14.24 | 0.98 | 122.24 | 224.09 |
| | 2022 | 55.53 | 57.56 | 13.61 | 13.85 | 0.96 | 114.39 | 208.89 |
| Amylose cont. | 2021 | 6.81 | 7.14 | 11.24 | 11.51 | 0.95 | 14.02 | 60.42 |
| | 2022 | 4.56 | 5.38 | 9.26 | 10.07 | 0.85 | 9.390 | 40.74 |

Note:
GV, genetic variance; PV, phenotypic variance; GCV, genotypic coefficient variation; PCV, phenotypic coefficient variance; h²b, heritability in broad sense; GA, genetic advance; GA%, genetic advance percentage. HD, heading date; PH, plant height; NTP, number of tillers/plant; PW, panicle weight; GYP, grain yield/plant; 1,000-GW, 1,000 grains weight.

influence on the expression of these traits by the environmental effect. Heritability in broad sense (*h2b*%) was measured for all recorded characterstics, *i.e.*, heading date (0.85), plant height (0.925), grain yield/plant (095), 1,000 grains weight (0.92), blast reaction (0.935), head rice (0.97) and amylose content (0.90). High hertability values up to the unity will reflect high genetic control of theses traits. Regarding genetic advance (GA), the topmost GA observed in GYP followed by head rice, PH, NTP and blast reaction. Otherwise, the minimal GA were reported by hulling and milling traits.

## Relationship among evaluated traits and genotypes

The interrelationship between the twenty lines and their background Sakha 104 with the measured agronomic attributes was explored through employing the principal component
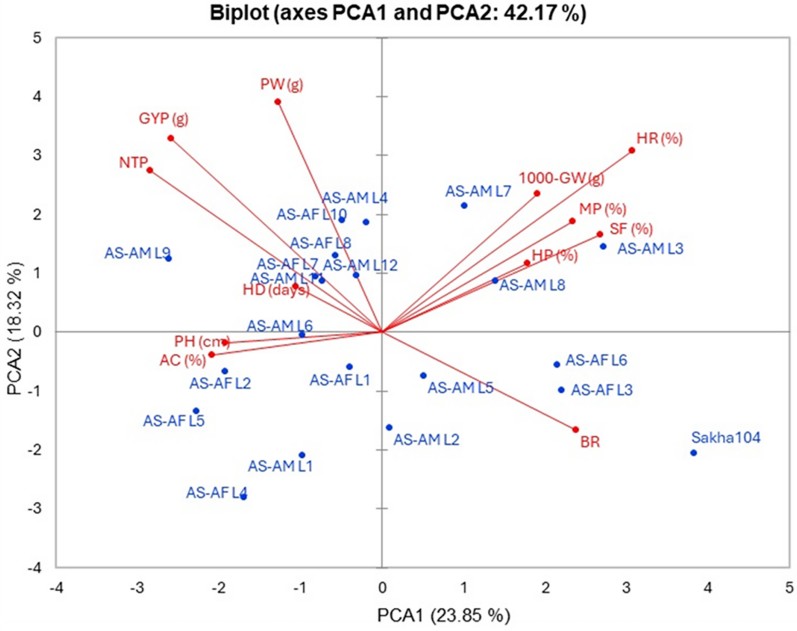

**Figure 5  PC-biplot for the evaluated traits and genotypes.** Days to heading (HD), plant height (PH), number of tillers/plant (NTP), panicle weight (PW), 1,000 grains weight (1000-GW), grain yield/plant (GYP), spikelet fertility (SF) and blast reaction (BR), hulling percentage (HP), milling percentage (MP), head rice (HR), and amylose content (AC).               

analysis (PCA). PCA1 and PCA2 describe a significant part of variance (23.85% and 18.32%, respectively). Those two PCAs were applied to initiate the PC-biplot as illustrated in Fig. 5. Moreover, PCA1 divided the genotypes into positive and negative sides, whereas PCA2 separated the measured traits into two sides (positive and negative). The genotypes located on the positive side of PCA1 had greater agronomic features, notably AS-AM L7, AS-AM L8 and AS-AM L3 lines, whereas genotypes existed in the negative side exhibited worse performance, such as AS-AF L4, AS-AF L5, and AS-AM L1 (Fig. 4). Likewise, measured variables with acute angle and situated on the positive side of PCA1, possessed strong and positive association such as 1000 grains weight, hulling percentage, milling percentage, head rice, and Spikelet fertility.

Similarly, heat map and hierarchical clustering of assessed traits and genotypes divided the evaluated traits into two groups (Fig. 6): the 1st group contained 1,000 grains weight (1,000-GW), head rice (HR), Spikelet fertility (SF), hulling percentage (HP) and milling percentage (MP), while the 2nd group categorized plant height (PH), heading days (HD), blast reaction score (BR), amylose content (AC), grain yield/plant (GYP), number of tillers/plant (NTP) and panicle weight (PW). In the same context, the genotypes separated into two main groups: group 1 involved six genotypes (AS-AF L6, AS-AF L3, AS-AM L8, AS-AM L4, AS-AM L3 and AS-AM L7), and group 2 had the rest of genotypes (14 genotypes), whereas Sakha 104 is considered out of groups. Finally, the first group of genotypes displayed superior values for traits 1,000-GW, HR, SF, HP, BR and MP (presented in blue color). Instead, the second genotypes group appeared good values in BR,

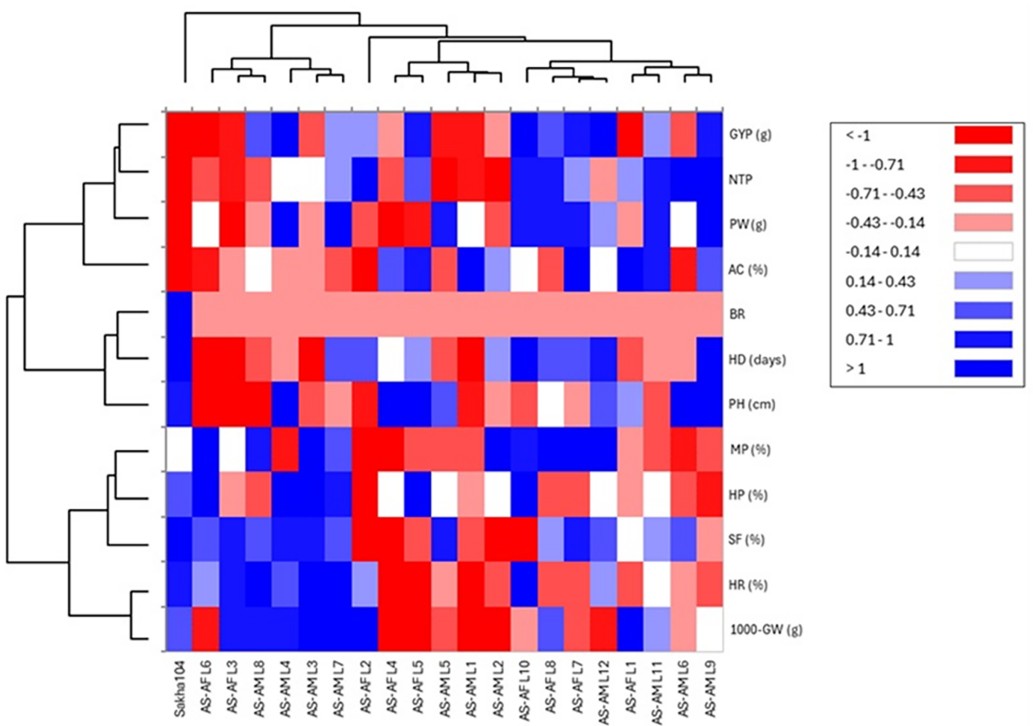

**Figure 6 Heatmap and hierarchical clustering divided the studied genotypes into two main groups based on evaluated agronomic characteristics.** Blue and red colors represent high and low values for the specific trait.

AC, GYP, NTP and PW. For plant height (PH), heading days (HD), and blast reaction (BR), negative values are desired (red color).

## DISCUSSION

Rice is a vital staple food that sustains over half of the global population, serving as a primary source of calories, carbohydrates, and essential nutrients for billions, particularly in Asia, Africa, and Latin America. There is a significant necessity for rice production to overcome and sustain global food security (*Brar & Khush, 2018*; *Gaikwad et al., 2021*; *Zhang et al., 2022b*). Mainly, traditional plant breeding and classical mendelian inheritance were primarily responsible for the increase in rice output. The family Gramineae (the genus *Oryza* belongs to this family) contains 24 species and two of them are the only cultivated species (*Oryza sativa* from Asian and *Oryza glaberrima* from African) in the world (*Garris et al., 2005*; *Zhang et al., 2022b*). *Oryza sativa* involves five major cultivar groups such as indica, aus, tropical japonica, temperate japonica and aromatic japonica (*Garris et al., 2005*). Rice production is usually impedance by several obstacles from biotic stresses such as diseases (blast, bacterial blight, sheath blight, rice yellow mottle virus, tungro virus, *etc.*) and insects (including stemborer, gall midge and plant hoppers). Furthermore, several abiotic stresses involving salinity, drought, heat and cold, in addition to soil toxicities will also affect the rice productivity (*Brar & Singh, 2011*; *Zhang & Xie, 2014*; *Sarma et al., 2023*).

The genetic diversity in rice and its wild relatives will enable it to adapt to a variety of environmental and climatic habitats. *Oryza* species comprised of eleven variable and distinguished genomes involving five allotetraploids genomes (BBCC, CCDD, KKLL, HHJJ, and HHKK) in addition to six diploid genomes (AA, BB, CC, EE, FF, and GG) (*Stein et al., 2018*). Notably, the rice wild relatives are categorized into three gene pools according to their phylogenetic relationship and ease of hybridization with cultivated rice (primary, secondary, and tertiary). The primary gene pool contains *O. sativa* (genome AA) that is closely related to cultivated varieties, the second gene pool involves *O. officinalis* (genomes BB to FF) with species having less relationship, whereas the third gene pool consists of *O. meyeriana*, *O. schlechteria*, and *O. ridleyi* (genomes GG to KKLL) with more distance from the cultivated rice (*Solis et al., 2020*; *Padmavathi et al., 2024*). For instance, the African farmed rice *O. glaberrima* (genome AA) originated from the wild parent *O. barthii*, whereas the Asian rice *O. sativa* was domesticated from its wild progenitor *O. rufipogon*. The Caribbean, South and Central America, and other new worlds are the origin of *O. glumaepatula*. It's one of the wild relatives of rice with diploid AA genome and can grow in riverbeds, waterlogged places, and marshes (*Sarla & Swamy, 2005*; *Thathapalli Prakash et al., 2023*). However, rice ancestors can be employed in breeding programs, subsequently, transferring traits related to biotic and abiotic stresses such as drought, salinity, heat responses and leaf blast resistance leading to improve existing rice varieties (*Thathapalli Prakash et al., 2023*). In addition, the pattern of inheritance in this trait is dominant in most cases and useful genes/QTLs could be exploit from wild rice species as a reservoir for rice enhancement. Techniques such as interspecific hybrids, chromosomal segmental substitution lines and alien introgression lines were performed and many genes/QTLs were transferred from wild species into cultivated rice (*Zhang & Xie, 2014*; *Xiao et al., 2019*; *Ramalingam et al., 2020*; *Ying et al., 2022*; *Kalboush et al., 2023*).

Blast in Rice, caused by the ascomycete fungus *Magnaporthe grisea* (*Pyricularia oryzae*), is regarded one of the most devastating disease that infect rice and Poaceae plants. It reduces rice output by 10-30% and up to 90% in ideal conditions (*Skamnioti & Gurr, 2009*; *Xiao et al., 2019*). Under Egyptian environment, blast disease reduced the rice output by 50% in the heavy infected seasons (*Kalboush et al., 2023*). In China (Guangdong province), around 10,000 hectares were affected by blast disease, with roughly 900 ha seriously damaged in 2016 (*Xiao et al., 2019*). Furthermore, from 74% to 100% losses in rice production was detected in India under extreme conditions (*Ramalingam et al., 2020*). Breeding for new blast-resistant cultivars is essential to overcoming blast sensitivity and yield loss. The most popular and effective technique for broadening the blast resistance gene pool in rice is to accumulate pyramid resistant (*R*) genes within a given cultivar. Additionally, understanding the mode of inheritance, gene type and stability of the resistant is essential to transfer and maintain blast resistance in cultivated types (*Deng et al., 2017*; *Xiao et al., 2019*). Aggregating numerous QTLs would imply various sources of partial resistance, which might reduce disease dispersion and maintain low selection pressure in the *P. oryzae* population, resulting in long-term resistance (*Nizolli, Pegoraro & de Oliveira, 2021*).

At the present study, the Egyptian cultivar Sakha 104 (released in 1999 based on developing from GZ 4096-8-l/GZ 4100-9-1), subjected to genetic improvement for yield, grain quality improvement and leaf blast resistance. Two crosses (Sakha 104 × *O. glaberrima* and Sakha 104 × *O. glumaepatula*) were performed and 20 promising genotypes were selected and evaluated up to F8 generation. Heading date (HD), plant height (PH), blast reaction, number of tillers/plant (NTP panicle weight (PW), grain yield/ plant (GYP), 1,000 grains weight (TGW), spikelet fertility (SF), milling percentage (MP), hulling percentage (HP), amylose content (AC), and head rice (HR)), were measured and recorded. Lines AS-AF L3 and AS-AF L6 presented the earliest heading performance, whereas AS-AF L6 and AS-AF L3 displayed the best values in plant height. Otherwise, Lines AS-AM L9 and AS-AM L6 manifested the uppermost number of tillers/plant whereas the best panicle weight was assigned to lines AS-AM L9 and AS-AML7, respectively. Furthermore, spikelet fertility, grain yield/plant and 1,000-grains weight were evaluated. The uppermost values for 1,000-grains weight were given by lines AS-AM L3 and AS-AML7. Sakha 104 recorded the best percentage in spikelet fertility, while the preferable values were assigned to lines AS-AM L4 in grain yield/plant. These findings are in consonance with those obtained by *Zhang et al. (2022b)*, who crossed 160 *O. sativa* accessions of rice and 170 of 7 AA genome species as a donor parents with three superior cultivars from *O. sativa* containing RD23 (an indica variety), Yundao 1 (a japonica variety), and Dianjingyou 1 (a japonica variety). Thus, they evaluated some agronomic traits from more than one species like plant height, 1,000-grain weight, glabrous hull, drought resistance, aerobic adaptation and blast resistance. *Futakuchi, Fofana & Sié (2008)*, *Ndjiondjop et al. (2018)*, *Brar & Khush (2018)* confirmed the ability of *Oryza glaberrima* to improve the lodging tolerance in *Oryza sativa*. The presented data in Figs. 2 and 3 revealed the phenotypic differences in earliness and plant height between the Egyptian background Sakha 104 and the derived lines from both African and American rice.

Grain yield, which is determined by spikelet fertility, grain weight, and panicle number, continues to be a top breeding goal. Grain Number 1a (GN1a) and Grain Yield 1 (GY1) are examples of QTLs that control grain number and panicle morphology. Breeders can use genomic selection (GS) and marker-assisted selection (MAS) to pyramid these loci into elite lines. For instance, the qGY2-1 QTL, which is associated with higher grain weight, has been introduced into well-known cultivars such as IR64, increasing yield by 15–20% (*Gouda et al., 2020*; *Li, Pan & Li, 2022*). *Gaikwad et al. (2014)* generated more than 2,000 lines from the wild species with AA genome. Seventeen hybrids were developed and assessed in large-size plots and four hybrids of these (ILH867, ILH299, ILH901, and ILH326), had significant heterosis and grain yield-related traits over the parental lines. Meanwhile, the cross among *O. sativa* (cultivar Dianjingyou 1) and *O. meridionalis* yielded three BC6F2 segregation populations. These three genotypes explored pleiotropic phenotypes on number of primary branches (NPB), grain number per panicle (GNPP), panicle length (PL), number of secondary branches (NSB), and grain width to enhance panicle architecture and yield potential.

The percentage of edible white rice that is extracted from harvested grains is determined by the hulling (husk removal) and milling (bran removal) percentages. Grain form, chalkiness, and husk thickness all affect these characteristics. Higher milling recovery is usually seen in cultivars with long, thin grains, such as Basmati varieties. Important candidates for genetic improvement include the genes GS3, which controls grain length, and Chalk5, which controls chalkiness. In the Japonica cultivar, CRISPR-Cas9 editing of Chalk5 increased milling yield by 40% by reducing chalkiness (*Anindya et al., 2019*). Notably, Reiho and Giza177 had the best values for milling percentage under 2 years of study (73.35, 72.66%), whereas for head rice, Reiho and Shin2 genotypes gave the highest percentages (69.42, 68.60) (*Al-Daej, 2022*; *Ali et al., 2023*). *Lu et al. (2023)* reported the milling and the nutritional quality of grains in the Indian variety Yuenongsimiao (YNSM). YNSM presented excellent quality and appearance with high gel consistency and low amylose contents whereas *Abd El-Aty et al. (2022)* crossed between eight rice genotypes and produced 28 F1. Sakha 106 × Sakha 108 and Sakha 106 × Sakha 107 crosses had the best and desirable values in head rice, milling (%), hulling (%), and the lowest amylose content under well-irrigation and drought conditions.

Environmental stresses like heat and drought have a significant impact on spikelet fertility, which is the percentage of filled grains per panicle. Up to 50% less fertility results from pollen sterility caused by high temperatures during blooming. Under stress, the fertility of thermotolerant cultivars is maintained by genes such as TT1 (thermotolerance 1) and qHTSF4 (QTL for heat-tolerant spikelet fertility). According to *Cheabu et al. (2019)* and *Kumar et al. (2021)*, the HT54 line maintained 80% spikelet fertility at 38 °C, while susceptible types only maintained 50%. Furthermore, balanced nutrition management improves grain fullness and pollen viability, especially when nitrogen and potassium are applied.

Regarding blast response, the value of leaf blast reaction for all examined lines in the current study, remained consistent during the two seasons of evaluation. Otherwise, the causal agent *Magnaporthe oryzae* infects rice leaves, stems, nodes, panicles, roots and attacks rice throughout the vegetative cycle. Once the fungus enters the plant cell, it multiplies rapidly and the visible blast symptoms will appear (*Wilson & Talbot, 2009*). Resistant (*R*) genes and defense response (DR) genes in rice play a vital role in the plant defense against fungal infection and contribute to broad-spectrum blast resistance (*Devanna, Vijayan & Sharma, 2014*; *Meng et al., 2020*; *Devanna et al., 2022*; *Biswas et al., 2023*). Remarkably, more than 100 *R* genes were discovered in the rice genome and 38 of them have been characterized and cloned. Eight genes of those 38 genes provide the broad-spectrum of blast resistance (*Singh et al., 2020*; *Devanna et al., 2022*). Blast-resistance QTLs firstly identified in the Moroberekan (the widely grown African variety) (*Khush & Jena, 2009*). Consequently, it is recommended to identify and characterize new QTLs with broad spectrum resistance against blast infection to enhance the cultivars resistance (*Devanna et al., 2022*).

Estimating heritability and genetic advance are extremely valuable markers of the desired traits after genotype selection. Genetic advance percentage was grouped in three categories as low (0–10%), moderate (10–20%) and high (≥20%). The highest GA was

observed in traits such as GYP, HR, PH, NTP and blast reaction. These findings are in harmony with those reported by *Roy & Shil (2020)* who found that heritability value in F3 lines (Tulaipanji × IR64 × PB1460), F5 lines (Tulaipanji × IR64), and F3 lines (Badshabhog × Swarna sub1) was highest in the trait grain length (99.97%) followed by grain number per panicle (99.77%), plant height (98%), and grain weight (99.29%), respectively. Likewise, genetic advance (GA) for the grain number per panicle trait was quite high (106.93) with genetic advance percentage 50.98. High heritability coupled with high genetic advance were detected in blast reaction, GYP, HR and PH, which may reflect the presence of additive gene action that control these traits and can be enhanced by simple selection (*Nihad et al., 2021*; *Acevedo-Siaca et al., 2021*; *Lu et al., 2023*; *Nivedha et al., 2024*).

The relationship among the assessed lines and the agronomic traits revealed some genotypes such as AS-AM L7 and AS-AM L8 that had positive and high agronomic characteristics based on PCA analysis. *Vaidya et al. (2024)* studied 120 recombinant inbred lines (RIL) and carried out the PCA for many agro-morphological traits. They found that the first five components counted 74.33% of the total variance and the uppermost positive eigen value observed in PC1 for panicle length and plant height, reflecting the maximum contribution to variation. Eventually, a total 64 genotyped were studied to determine the variability in 18 agro-morphological traits in Rwanda. About 72% of total variability were assigned to seven components based on the PCA analysis (*Mvuyekure, 2018*).

## CONCLUSIONS

Biotic stressors, such as blast infection, minimize rice output annually and threaten world crops. To maintain food security, blast resistance is being actively bred into new cultivars. The current study produced and assessed 20 genotypes with their background parent (Sakha 104) under both normal and blast conditions. The desired genotypes were evaluated for days to heading (HD, plant height (PH), number of tillers/plant (NTP), panicle weight (PW), 1,000 grains weight (TGW), grain yield/plant (GYP), Spikelet fertility (SF), blast reaction (BR), hulling percentage (HP), milling percentage (MP), head rice (HR) and amylose content (AC)). In general, the most derived lines have proven superior than the genetic background parent. There was no line with superiority in all studied traits. The African lines had shorter structure, early heading and blast resistance while maintaining strong yield potential (AS-AF L3 and AS-AF L6). On the other hand, the American lines demonstrated the best yield potential, yield components, excellent grain quality and blast resistance (AS-AM L3, AS-AM L4 and AS-AM L9). From here, crossing American and African derived lines might produce new and superior rice genotypes.

### Funding

This work was supported by the Deanship of Graduate Studies and Scientific Research at Qassim University with financial support (QU-APC-2025). The funders had no role in

study design, data collection and analysis, decision to publish, or preparation of the manuscript.

## Grant Disclosures
The following grant information was disclosed by the authors:
Deanship of Graduate Studies and Scientific Research at Qassim University: QU-APC-2025.

## Competing Interests
The authors declare that they have no competing interests.

## Author Contributions
- Walid H. Elgamal conceived and designed the experiments, performed the experiments, analyzed the data, prepared figures and/or tables, authored or reviewed drafts of the article, and approved the final draft.
- Mostafa M. Elshenawy conceived and designed the experiments, performed the experiments, authored or reviewed drafts of the article, and approved the final draft.
- Samah M. Abdelkhalek conceived and designed the experiments, analyzed the data, prepared figures and/or tables, authored or reviewed drafts of the article, and approved the final draft.
- Dayun Tao conceived and designed the experiments, prepared figures and/or tables, authored or reviewed drafts of the article, and approved the final draft.
- Jiawu Zhou conceived and designed the experiments, performed the experiments, prepared figures and/or tables, authored or reviewed drafts of the article, and approved the final draft.
- Jing Li conceived and designed the experiments, performed the experiments, analyzed the data, authored or reviewed drafts of the article, and approved the final draft.
- Medhat Rehan conceived and designed the experiments, analyzed the data, prepared figures and/or tables, authored or reviewed drafts of the article, and approved the final draft.

## Data Availability
The raw data are available in the Supplemental Files.

## Supplemental Information
Supplemental information for this article can be found online at http://dx.doi.org/10.7717/peerj.19453#supplemental-information.

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
