# Peer review of "Improving the cultivated rice Sakha104 (Oryza sativa L.) using gene pools of some relative wild species"

_PeerJ, doi:10.7717/peerj.19453_

## Round 0.1 · original submission · Minor Revisions

The manuscript titled "Improving the cultivated rice Sahka104 (Oryza sativa L.) using gene pools of some relative wild species" represents a significant effort to enhance rice productivity and blast resistance by utilizing wild rice relatives. While the study provides valuable insights into genetic improvement, several issues require clarification and revision to enhance the manuscript’s clarity, accuracy, and scientific rigor.

Please check the comments of both reviewers, particularly revising figure panels and considering title changes. Please clarify how you isolated the control blocks from the infected blocks, given the airborne nature of the disease.

You do not have to perform new experiments, but you should improve the discussion following the comments of the reviewers

Sincerely

Reviewer 1 ·

Basic reporting

1- The language was good
2-The general preparation was good, however there are too much repetition in the discussion, I would suggest to rewrite and summarize the results and discussion
3- The references were good
4- The title make me confused about the target, is it general improving, or they looking for blast resistance genes ?

Experimental design

1- Technical issues they did not mention, how did they separate the control from the infected blocks especially this disease can transfer by air.
2- They used natural infection which is not right. This is big mistake as they do not have an equal inoculum for each experiment block. This is not scientific way to compare between treatments they should use same dose for all.
3- Systemic chemical, that they use for control, this could interfere with metabolic pathways of the plant which increased or decreased the gap between treatments
4- Please clarify they used 50 cm row and 5 meter why? what a bout plant density ?
5- Before doing this experiment, have they check the wild relative whether if they have the resistance genes to their local pathogen strain or not ?

Validity of the findings

1- In the rows 373-381 you mention 8 major genes, did they investigate them using molecular methods ?
2- This type of work would be accepted in 80s of the last century, but now we have much more modern methods as a check points to follow the interested traits, for example conventional PCR, RT-PCR or even NGS to have overall view about what has happened and genome rearrangement

Additional comments

I would like to thank you for chosen me as a reviewer.

Reviewer 2 ·

Basic reporting

Line 154: standard evaluation system
Fig 1: The location of the F3 and F4 generations planted in China seems to be incorrect. Please correct it.
Line 225: “latest values in blast reaction”, please correct it.
Fig 3 & 4: It is better to mention the minimum value line and maximum value line within brackets.

Experimental design

Methods section clearly written.

Validity of the findings

In general, the discussion section is poorly written. Need substantial improvements in this section.

Lines 267 & 268: Just repeating the introduction. Please rewrite these two lines.
Lines 270 & 271: Please rewrite.
Line 275: Please provide a reference
Line 282: Better to change the “devastating factor” to devastating disease or fungi
Line 284: Please rewrite.
Line 288 & 289: Please rewrite.
Line 293: cultivated cultivar is just a repetition. Cultivated types are simply referred to as cultivars. Therefore, remove the word cultivated.
Line 294 & 295: Please reword.
Line 298-299: Unclear.
Line 303: better to write as cultivated rice instead of just rice.
Does O. glaberrima is also consist of AA genome? Please mention it in line 316.
Line 317-318: unclear.
Line 317,18 & 19: need citations
The paragraph from the Line 349-364 seems not focused. Please rewrite.
Line 369: Resistant genes
Line 376: Not clear! More than 500 Blast-resistant QTLs identified?
Throughout the discussion, give the citations correctly.
Need to improve the grammar, especially in the discussion section.

Additional comments

Good job done with the research study. However, the manuscript needs improvements.

---

## Round 0.2 · Minor Revisions

Dear Dr. Rehan,

Some minor issues to be corrected/clarified before this is ready:

+++
Can the authors please clarify the crossing scheme depicted in Figure 1? What crosses were done to go from F1 to F8? Selfing? So, on average each F8 had 50% genetic contribution from the wild parent? At what stage (s) was phenotypic selection made? When were the 20 lines chosen? From how many plants?

+++
Figure 3C and 4A: the reader will want to compare differences between lines in specific traits. It is hard to do with this organization. Please re organize so that all of the bars for a particular trait are next to each other.

+++
Figure 6. How have the traits been normalized to make this plot? What are the numbers?

+++
Line 187. Reached...XX% of what?

It would be good if the paper was carefully read and edited for word choice, spelling, etc., before resubmission.

**Language Note:** The Academic Editor has identified that the English language must be improved. PeerJ can provide language editing services - please contact us at [email protected] for pricing (be sure to provide your manuscript number and title). Alternatively, you should make your own arrangements to improve the language quality and provide details in your response letter. – PeerJ Staff

Reviewer 2 ·

Basic reporting

The article was substantially improved.

Experimental design

I am satisfied with the materials and methods section

Validity of the findings

As requested by me during the previous review, the discussion section was sufficiently improved.

Additional comments

I am satisfied with the attempts made to improve the manuscript.

---

## Round 0.3 · Minor Revisions

Dear Dr. Rehan
Thanks for your new version. Before the manuscript is accepted, we need a new
round of revision addressing the points below.


"Some minor issues to be corrected/clarified before this is ready:
+++
can the authors please clarify the crossing scheme depicted in Figure 1? What crosses were done to go from F1 to F8? Selfing? So on average each F8 had 50% genetic contribution from the wild parent? At what stage (s) was phenotypic selection made? When were the 20 lines chosen? From how many plants?
+++
Figure 3C and 4A: the reader will want to compare differences between lines in specific traits. It is hard to do with this organization. Please re organize so that all of the bars for a particular trait are next to each other.
+++
Figure 6. How have the traits been normalized to make this plot? What are the numbers?
+++
Line 187. Reached...XX% of what?

---

## Round 0.4 · accepted · Accept

Congratulations on the acceptance of your manuscript.